# Efficient Pre-Training of LLMs via Topology-Aware Communication Alignment on More Than 9600 GPUs

**Guoliang He**[*†1], **Youhe Jiang**[*‡1], **Wencong Xiao**[2], **Kaihua Jiang**[2], **Shuguang Wang**[2],
**Jun Wang**[2], **Zixian Du**[2], **Zhuo Jiang**[2], **Xinlei Zhang**[2], **Binhang Yuan**[3], **Eiko Yoneki**[1]

[1]University of Cambridge, [2]ByteDance Seed, [3]HKUST

{gh512, yj367}@cam.ac.uk, biyuan@ust.hk, eiko.yoneki@cl.cam.ac.uk,
{hanli.hl, jiangkaihua, wangshuguang, wangjun.289}@bytedance.com,
{duzixian, jiangzhuo.cs, zhangxinlei.123}@bytedance.com

## Abstract

The scaling law for large language models (LLMs) depicts that the path towards machine intelligence necessitates training at large scale. Thus, companies continuously build large-scale GPU clusters, and launch training jobs that span over thousands of computing nodes. However, LLM pre-training presents unique challenges due to its complex communication patterns, where GPUs exchange data in sparse yet high-volume bursts within specific groups. Inefficient resource scheduling exacerbates bandwidth contention, leading to suboptimal training performance. This paper presents Arnold, a scheduling system summarizing our experience to effectively align LLM communication patterns with data center topology at scale. An in-depth characteristic study is performed to identify the impact of physical network topology to LLM pre-training jobs. Based on the insights, we develop a scheduling algorithm to effectively align communication patterns with the physical network topology in modern data centers. Through simulation experiments, we show the effectiveness of our algorithm in reducing the maximum spread of communication groups by up to $1.67$x. In production training, our scheduling system improves the end-to-end performance by $10.6\%$ when training with more than 9600 GPUs, a significant improvement for our training pipeline.

## 1 Introduction

Pre-training large language models (LLMs) at scale is a highly resource-intensive process that requires vast computational infrastructure. The performance of LLM training is fundamentally dependent on three factors: dataset size, computational power, and model parameters [19]. To meet these demands, companies continually enhance their computing infrastructure by incorporating cutting-edge GPUs and redesigning network architectures [14, 32, 40]. However, LLM pre-training presents unique challenges that distinguish it from conventional deep learning tasks — in this paper, we explore *how to develop an efficient resource scheduling mechanism to support the LLM training workflow to accommodate the resource-intensive and complex communication patterns in modern data centers*.

LLM pre-training is an exceptionally resource-intensive process. Given the pressing need to commercialize LLMs swiftly, accelerating the training process is paramount. However, training these models often spans weeks, requiring the deployment of thousands of GPU nodes per run. The ability to efficiently schedule and allocate resources is critical for both performance optimization and cost

---

[*]Equal contribution.

[†]Work done during internship at ByteDance Seed.

[‡]Correspondence to: Youhe Jiang <yj367@cam.ac.uk>.

39th Conference on Neural Information Processing Systems (NeurIPS 2025).

management. Furthermore, the unique data transmission patterns in LLM training — wherein GPUs communicate sparsely but at a high volume within specific groups — pose an additional challenge in leveraging modern multi-tier network topologies effectively.

Existing cluster schedulers (e.g., [42, 43, 44, 45, 7, 5]) fail to integrate network topology-aware scheduling specific to LLM workloads. The primary limitation is their lack of awareness of the high-volume, yet sparsely active distributed communication patterns inherent in LLM training. For example, Figure 1a indicates that 30% - 50% of the time is spent on communication during production LLM training, but studies [40] show that more than $99\%$ of the GPU pairs do not exhibit direct traffic, with data exchange occurring exclusively within specific communication groups, as shown in Figure 1b. Meanwhile, modern GPU clusters use multi-tier, fat-tree network topologies [1] (Figure 2b), and inefficient job placement leads to significant bandwidth loss and communication overhead. Current schedulers are not designed to optimize network-aware placement at the scale required by LLM pre-training jobs (LPJs).

To enable effective scheduling of LPJs in data centers, we identify two key challenges that limit the effectiveness of existing cluster schedulers.

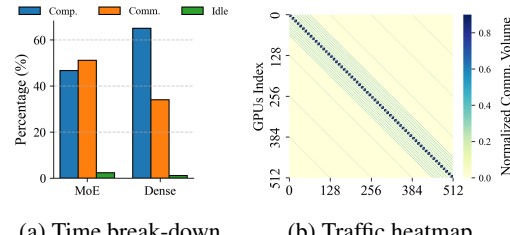

(a) Time break-down.        (b) Traffic heatmap.

Figure 1: Communication characteristics of LLMs training in production.

- **Misalignment of communication patterns with data center topology.** Schedulers optimize GPU locality through bin-packing but lack awareness of LPJ communication structures. Consequently, jobs may experience inefficient cross-switch communication despite being tightly packed.

- **Unaddressed trade-offs across communication dimensions.** There is a fundamental trade-off between aligning data parallel (DP) and pipeline parallel (PP) communication patterns. Since GPUs participate in both groups, perfect alignment for both is unachievable, and schedulers must carefully balance the two during placement.

To address the challenges, we present Arnold, a system that co-designs training frameworks and cluster scheduling, effectively aligning LPJs with modern data center network topology. To optimize training performance, we performed an in-depth characterization study to investigate the impact of physical network topology on LLM training. Based on the observation, we devise a scheduling algorithm to reduce the maximum weighted spread of communication groups for LPJs. We also develop a resource management policy that manages job queues to reserve nodes for imminent LPJs.

Through trace-based experiments, we show the effectiveness of our scheduling algorithm by benchmarking against other SOTA algorithms. We also perform a production run with 9600+ GPUs and show our proposed system improves the end-to-end training performance by $10.6\%$. In summary, our contributions include the following:

**Contribution 1.** We identify the challenge of aligning LLM communication patterns with modern data center topology for large-scale pre-training. We characterize the impact of physical network topology on individual communication operations and end-to-end training performance in modern data centers.

**Contribution 2.** We design a scheduling algorithm to effectively align communication patterns to the topology of the data center for LPJs, and a resource management policy to reserve nodes for the placement.

**Contribution 3.** Through comprehensive simulation experiments, we evaluate the effectiveness of the scheduling algorithm in reducing the maximum spread of communication groups by up to $1.67$x. In a production run, we verify the proposed scheduler can improve a 9600+ GPUs LPJ by $10.6\%$.

## 2 Background

**Distributed training.** LLMs are billion-parameter transformer-based models that must be trained with multi-GPU systems[39, 35]. Common training frameworks [36, 34] employ hybrid parallelization strategies to parallelize and accelerate the training process, including:

- Data parallelism (DP). The Zero Redundancy Optimizer (ZeRO) [33] shards model weights and gradients across data parallel processes and performs synchronization at the end of a training step by all-gather and reduce-scatter communication operations.

- Pipeline parallelism (PP). The layers of models are divided into several stages (PP size), and each stage interleave communication to adjacent stages as well as the computation within the stages. Inter-stage communication is performed by P2P communication operation like send-recv.

- Tensor parallelism (TP). Model weights within an PP stage are further sharded across multi-GPUs to alleviate the memory pressure. All-gather and reduce-scatter are necessary to synchronize the intermediate activation during forward pass and backward pass.

The combination of parallelism, i.e. hybrid parallelism, forms diverse communication patterns for GPUs, and training frameworks use communication group to manage the complexity. Each GPU is assigned to a DP, TP, and PP communication group at initialization. The illustrations of different parallelisms and communication groups are demonstrated in Figure 2a. Other parallelism strategies, such as sequence parallelism and expert parallelism, are excluded from discussion as they are for specific scenarios (long context and MoE models respectively).

Previous works [24, 36] have identified that TP communication groups should be prioritized to GPUs located within the same node to utilize the high-bandwidth NVLink interconnection due to stringent data dependencies. Thus, the inter-node communication only takes place within the DP and PP groups. As only inter-node communication is sensitive to physical network topology, the communication patterns of DP group and PP groups are the focus of this paper.

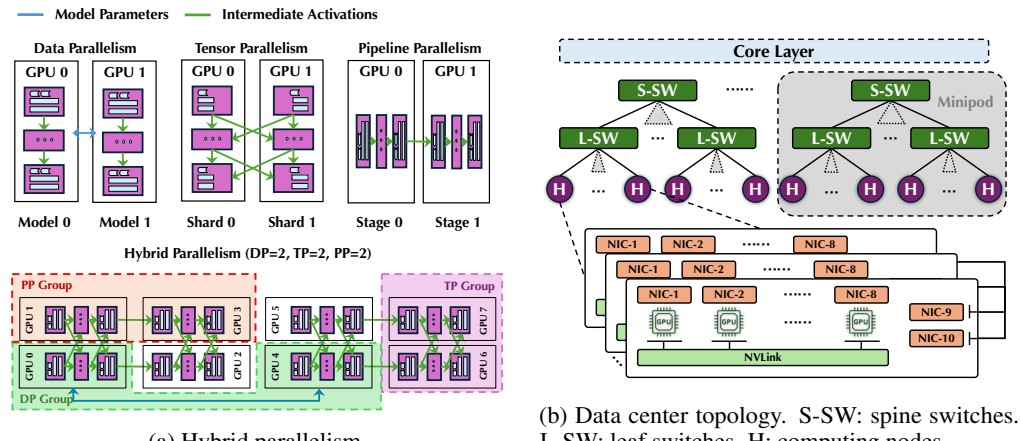

(a) Hybrid parallelism.

(b) Data center topology. S-SW: spine switches. L-SW: leaf switches. H: computing nodes.

Figure 2: LLMs parallelism and data center topology.

**Data center topology.** Figure 2b gives an overview of our HPC cluster, which is similar to other modern data centers. More than 2000 nodes are interconnected by three layers of switches, forming a CLOS-like topology [8]. The leaf switch (L-SW) is denoted as s0, which interconnects nodes within the same rack. Then, several s0 switches link to a spine switch (s1), forming a minipod of nodes. Finally, s1 switches link to core switches, enabling communication between minipods. The switches in each layer have 32 ports both for uplinks and downlinks. The greatest number of hops occurs when the nodes of two different minipods communicate with each other. The compute nodes are equipped with 8 H800 GPUs, each of which is connected to an InfiniBand [26] NIC. GPUs within a node are connected by high-bandwidth links such as NVLink [29] with a bandwidth of 400GB/s, while inter-node communication is achieved via the InfiniBand network (400GB/s).

## 3 Observation and Challenges

Given a user-specified number of GPUs and degree of hybrid parallelism of an LPJ, job scheduling systems enqueue the job and perform resource scheduling to find the best placement in GPU clusters. However, we observe existing scheduling systems fail to align LLM communication patterns with data center topology in practice.

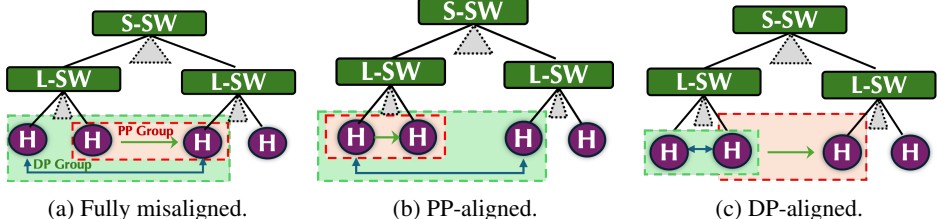

(a) Fully misaligned.  (b) PP-aligned.  (c) DP-aligned.

Figure 3: Alignment of communication patterns. One DP group and PP group is highlighted.

***Observation 1: Misalignment of job placement results in increased cross-switch communication.***
SOTA cluster schedulers [42, 2, 44] apply a bin-packing strategy to enhance GPU locality of LPJs. However, as shown in Figure 3a, even if the scheduler packs an 4-node (32 GPUs) LPJ inside a minipod, the communication groups may still not be aligned, because both DP and PP groups engage cross-spine-switch communication that has a longer distance. This misalignment stems from the scheduler's lack of awareness of the LPJ's communication structure at scheduling time, limiting its ability to allocate GPU resources according to the job's communication patterns.

***Observation 2: Unresolved trade-offs between DP and PP communication priorities.*** Figure 3b and 3c show two potential alignments of the LPJ, with one prioritizing DP communication and the other prioritizing PP. This presents a fundamental trade-off between the two, because DP and PP are orthogonal parallelism strategies widely used in LLM training. Each GPU participates in both a DP and an PP group, making it impossible to perfectly align both communication patterns simultaneously. A well-designed scheduler must consider this trade-off and balance the alignment needs of both group types during job placement.

***Challenge: Communication and topology-aligned scheduling for LPJs.*** To effectively schedule LPJs, the scheduler must be aware of the diverse communication patterns and minimize their spread in data centers. Furthermore, effective balance between the spread of DP and PP groups is critical, which requires in-depth characterization of communication patterns in modern data centers.

## 4 Characterization of Communication Patterns for LPJs

Although prior studies [44, 2] have explored locality and topology, their scope is constrained by (1) a focus on data-parallel (all-reduce) workloads and (2) limited consideration of inter-node topology. To address these gaps, we conduct NCCL tests, a benchmarking suite designed to measure the latency and bandwidth of communication operations used by NCCL [27, 28], to study the impact of inter-node topology. We focus on inter-node topology across minipods, as the scale of LPJs typically necessitates allocating computing nodes across multiple minipods, where the slowest communication path often dictates the overall communication overhead.

**Communication operation performance.** Figure 4 studies the performance of communication operations. We use the bus bandwidth (BusBw) as a performance measurement, which reflects the peak hardware bandwidth by accounting for the number of ranks for collective communication.

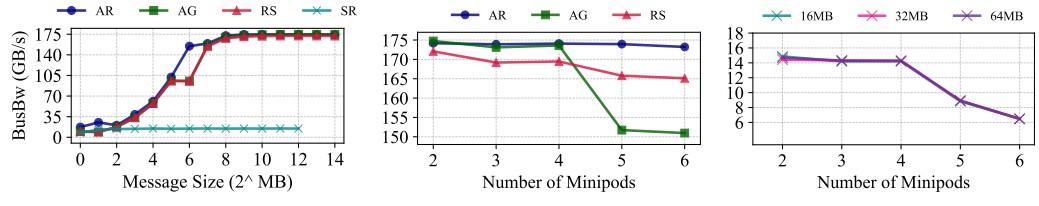

(a) BusBw over message sizes.  (b) BusBw of collective operations.  (c) BusBw of send-recv.

Figure 4: Performance of communication operation. (AR: all-reduce, AG: all-gather, RS: reduce-scatter, SR: send-recv)

Figure 4a measures the inter-minipod communication bandwidth and illustrates the performance of different communication patterns over message sizes. For collective communication, the message size must be larger than $2^8$ ($\approx 256$) megabytes to fully utilize the bandwidth, while for P2P communication like send-recv, a small message size ($\approx 2$ megabytes) can saturate the bandwidth. Using a widely

adopted analytical model detailed in Appendix B and substituting the parameters with a 7B GPT-based model, we can obtain that the data volumes of the DP and PP groups are 2 GB and 30 MB respectively, indicating that the bandwidth is fully utilized.

The degradation in BusBw by expanding communication groups across minipods is illustrated in Figures 4b and 4c. Performance decreases by up to 17% for collective operations and 70% for the P2P operation as communication extends over additional minipods, highlighting the critical importance of GPU locality and alignment. Additionally, our findings suggest that co-located jobs may experience reduced bandwidth contention in multi-tenant cluster environments, as evidenced by the inter-job interference patterns documented in Appendix C.

**End-to-end training performance.** Based on the characterization of individual communication operations, we proportionally down-scaled a production model and ran the workload with 96 GPUs, spanning 2 minipods, to further understand the impact of network topology on LLM training.

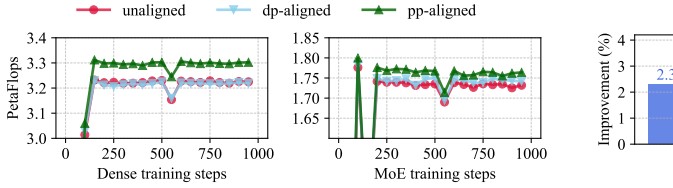

(a) Comparison of three different placement strategies for two types of LLMs.

(b) Performance improvement by scaling model sizes under the optimized alignment.

Figure 5: End-to-end training performance.

Figure 5a shows the throughput of the LPJ under the three different placement strategies. The dp-aligned placement is illustrated in Figure 12. The throughput becomes stable after 200 steps except for a slight fluctuation around the 550th step due to garbage collection. PP-aligned placement consistently outperforms the other two, demonstrating that prioritizing PP group communication leads to improved performance. The average improvement for the dense model and the MoE model is 2.3% and 1.8% respectively. For the dense model, we also observe that the PP communication dominates the communication overhead, since prioritizing the placement of DP groups leads to no speedup. For the MoE model, reducing the spread of both the DP and PP groups contributes to performance gains, with the optimizations of the PP group providing more improvements.

Figure 5b shows that if we scale the model size by adding more layers, the performance improvement continues to increase. We attribute this to communication being the primary performance bottleneck, with larger models further amplifying communication volume. Thus, more pronounced benefits are obtained as the model size increases.

We further examine the sensitivity of training performance to intra-minipod network topology by varying node placement within a single minipod. For a dense 24B parameter model, the maximum observed performance variation is 0.3%, and the impact is negligible for other models. Since the communication overhead of a group is typically dominated by the slowest link, and LPJ communication groups frequently span multiple minipods, we conclude that training performance is largely insensitive to intra-minipod topology.

We repeated the characterization experiment in another GPU cluster detailed in Appendix D, and found that the best placement can be subjected to model sizes and GPU types. Since LPJs are typically scheduled in advance and deployed for a long duration, it is essential to perform a characterization beforehand to identify communication bottlenecks within the group. This enables the optimization of placement strategies accordingly and balances the trade-off, as detailed in §5.

## 5   Scheduling Algorithm

The core component of Arnold is its scheduling module, which is designed to effectively allocate GPUs to LPJs based on the user-specified number of GPUs and the degree of hybrid parallelism. In this section, we formally define the scheduling problem and present our proposed solution.

## 5.1 Workload Representation

Arnold represents an LPJ by a communication matrix, where a row represents a PP group and a column represents a DP group. Formally, given a job specification including the total number of GPUs, the degree of PP, TP, and then Arnold computes the size of the communication matrix using Equation 1.

$$DP = \#GPUs/TP/PP$$
$$\#row = DP/(8/TP) \qquad (1)$$
$$\#col = PP$$

An example of 96 GPUs and $DP = 6, PP = 2$ is shown in Figure 12. For a node $v_{ij}$ in the communication matrix, it is attached with a vector $[v_w, v_d, v_p]$, representing the size of weight, the DP and PP communication volume per GPU respectively. Those values are computed using the analytical model in Appendix B, and is used to balance the trade-off between DP and PP groups.

## 5.2 Objectives

The scheduling objective function aims to minimize the maximum spread for both DP and PP groups. For a node $v_{ij}$ in the communication matrix, the number of possible assignments to a minipod is $k$. Therefore, it has a one-hot decision vector $x_{ij}$ of length $k$, representing the decision to place to one of the $k$ minipods. Then the objective function can be written as Equation 2.

$$\textit{MIN} \quad [\alpha \max_i(D(x_{i1}, x_{i2}, ...x_{in})) + \beta \max_j(D(x_{1j}, x_{2j}, ..., x_{mj}))] \qquad (2)$$

Where, the distance $D$ between the $n$ vectors is defined as follows:

$$D(v_1, v_2, \ldots, v_n) = |\{i : \exists j, l \in \{1, 2, \ldots, n\}, j \neq l, v_j[i] \neq v_l[i]\}| \qquad (3)$$

Intuitively, the distance measures the spread of a communication group, i.e. if any two vectors differ in the $i$th position, the $i$th position adds one to the distance. The objective function aims to minimize the weighted sum of the maximum spread of DP and PP groups, and the maximum is taken because the slowest communication group is the straggler to slow down the whole training process. The weight parameters $\alpha$ and $\beta$ represent the affinity that controls the trade-off between DP and PP groups, and $\alpha + \beta = 1$. Equation 2 cannot be solved by off-the-shelf solvers efficiently for online scheduling due to the discrete distance calculation, and we seek for simplifications.

**Domain-specific simplification.** We identify that communication groups are homogeneous and synchronous for LPJs because nodes are gang-scheduled and must synchronize their gradients at the end of a training step. As a result, each PP group always starts approximately at the same time and performs the same amount of computation and communication. Similarly, DP groups perform gradient synchronization at the same time. The characteristics allow us to simplify the scheduling objective function by coarsening a scheduling unit as a communication group. We therefore transform Equation 2 into a bin-packing-like formulation:

$$\textit{MIN} \quad [\alpha \sum_j (y_j) + \beta T] \qquad (4)$$

$$\text{s.t.} \quad \forall i : \sum_j s_{ij} \leq T \quad \textbf{(Max Spread)} \qquad (5)$$

$$\forall j : \sum_i p_{ij} \leq c_j y_j \quad \textbf{(Capacity Const.)} \qquad (6)$$

$$\forall i : \sum_j p_{ij} = 1 \quad \textbf{(Allocation Const.)} \qquad (7)$$

$$\forall i, j : p_{ij} \leq s_{ij} \quad \textbf{(Minipod Selection)} \qquad (8)$$

$$\forall j : y_j \in \{0, 1\} \qquad (9)$$

$$\forall i, j : s_{ij} \in \{0, 1\}, p_{ij} \in [0, 1] \qquad (10)$$

Where $y_j$ indicates whether the $j$-th minipod is used and $c_j$ is the normalized capacity of the minipod, updated dynamically based on the number of available nodes. $s_{ij}$ denotes whether the $i$-th communication group is allocated to minipod $j$, $p_{ij}$ denotes the percentage of the $i$-th communication group allocated to minipod $j$. $T$ is an introduced auxiliary variable that allows us to minimize the maximum spread of communication groups. $\alpha$ and $\beta$ are the affinity parameters as before. Overall, minimizing $T$ effectively consolidates communication groups into the smallest possible number of minipods, while the objective term $\sum_j(y_j)$ controls the spread of the other communication group.

**A simple example.** We consider the PP group as a scheduling unit. By setting $\alpha = 0$, the scheduler can place each PP group into a minipod, causing more cross-switch communication for DP groups,

while by setting $\beta = 0$, the scheduler minimizes the overall usage of minipods, although the placement may cause cross-switch communication for PP groups. Together, Equation (4) is the balanced optimization objective function. Equation (5) restricts the maximum spread of PP groups to be less than or equal to $T$. Equation (6) restricts that the allocation cannot exceed the capacity of a minipod. Equation (7) suggests that the sum of the allocation percentage must be equal to 1 for every PP group. Equation (8) suggests that if a minipod has some percentage of a PP group, then it is considered used. Finally, Equation (9) and (10) define the range of the binary variable $y_j$, $s_{ij}$ and the continuous variable $p_{ij}$. As a concrete example, the values for those variables in a simulated experiment are listed in Appendix H.

In this formulation, all variables are either an integer or a fraction. Therefore, the objective function can be solved using off-the-shelf mixed-integer programming (MIP) solvers efficiently for online scheduling [4]. After solving the MIP, we assign continuous rank indices to nodes belonging to the same minipod to reduce cross-switch communication within each communication group.

**Balancing the trade-off.** The affinity parameters in Equation 4 require balance of the trade-off between the DP and PP groups, which depends on the model configurations and GPU types (§4). To perform online scheduling, we store the characterization results in §4 to a database, and we search for the best match to determine the values of the affinity parameters. The communication matrix computes the per-GPU parameters ($v_w$) and communication volumes ($v_d, v_p$). We then compute the average ratio of computation-to-communication and DP-to-PP volume as $r_1 = \frac{mb \times v_w}{v_d + v_p}$ and $r_2 = \frac{v_d}{v_p}$, where $mb$ is the size of the microbatch. These ratios are used to find the best matching job in the database by comparing the Euclidean distance, i.e. $MIN_{r_i, r_j} \sqrt{(r_1 - r_i)^2 + (r_2 - r_j)^2}$, because GPUs exhibit comparable performance characteristics if they have similar computational load and communication volume.

LPJs are associated with metadata $\langle GPU_{type}, j_{dp}, j_{pp} \rangle$, where $j_{dp}, j_{pp}$ corresponds to the improvement of DP-aligned, and PP-aligned placement strategies. The affinity parameters $\alpha$ and $\beta$ are then derived based on the relative performance improvement of $j_{dp}$ and $j_{pp}$, i.e. $\alpha = \frac{j_{dp}}{j_{dp} + j_{pp}}$ and $\beta = \frac{j_{pp}}{j_{dp} + j_{pp}}$.

Due to the importance of LLM training and their unified architectures, LPJs are scheduled in advanced and pre-characterized, so the profiling data in the database can be looked up in online scheduling. For example, for a 24b dense model in the H800 GPU cluster, the scheduling unit is set to the PP group and $\alpha$ is set to zero as PP groups clearly dominate the communication overhead. For the 24b MoE model, $\alpha = 0.3$ and $\beta = 0.7$.

### 5.3 Resource Management

The scheduling algorithm computes a globally optimal placement for LPJs in the GPU cluster, which inevitably conflicts with other jobs. To address this, we develop a queuing policy to manage the job queue and reserve resources for the imminent LPJ.

Algorithm 1 illustrates our scheduling policy. Once the LPJ is planned, the scheduler solves the MIP equation and reserves the resources. Since then, incoming jobs are preferentially allocated outside the reserved zone. Otherwise, to improve resource utilization, if the predicted job completion time (JCT) of an incoming job precedes the arrival time of the LPJ, it may still be scheduled within the reserved zone. If neither of the conditions can be satisfied, the job is delayed in the scheduling interval. We also employ an ML-driven JCT predictor to balance the trade-off of queuing delay and resource utilization. The setup and evaluation are detailed in Appendix F and G respectively.

## 6  System Implementation

We have implemented a prototype of Arnold with more than 3k lines of Python codes. The prototype consists of the scheduling module and a trace-driven simulator that can replay production traces. We also have a version of the deployment integrated with Kubernetes [21]. For training frameworks, we build on top of Megatron [24] and modify it to ensure that communication groups follow the placement provided by Arnold. Figure 6 gives an overview of Arnold.

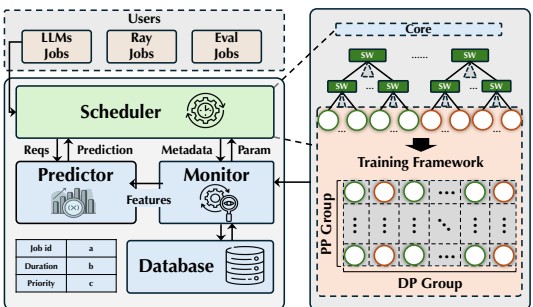

Figure 6: Architecture overview of Arnold.

# 7 Evaluation

We evaluate Arnold using both simulation and real-cluster experiments. To benchmark scheduling algorithms cost-effectively, we develop a simulator, as direct evaluation on production clusters is prohibitively expensive. After identifying the highest-performing scheduling algorithm through simulation, we deploy it on our production cluster to validate its effectiveness under real workloads.

## 7.1 Simulation Experiments

**Baselines.** We compare the scheduling algorithm with the following baselines.

1. Best-fit [31] assigns the nodes to the minipod with the least remaining resources.

2. Random-fit [43] assigns nodes to minipods randomly such that the assignment is balanced and fair.

3. GPU-packing [42, 44] is an effective placement strategy applied by state-of-the-art GPU cluster schedulers that pack multiple jobs to the same GPU. We modify the algorithm to pack multi-GPU jobs to a minipod to satisfy the network topology semantics.

4. Topo-aware [2] is a GPU topology-aware placement algorithm. It represents the workload as a job graph (similar to our communication matrix) and the topology as a physical graph. Then it recursively bi-partitions the physical graph and maps the job graph to the sub-graphs (Hierarchical Static Mapping Dual Recursive Bi-partitioning [10]). The graph bi-partitioning is implemented by the Fiduccia Mattheyses algorithm [11].

**Metrics.** The weighted sum of the maximum spread as in Equation 2 is used to evaluate scheduling algorithms. To evaluate scalability, we measure the scheduling latency.

Table 1: Benchmark setting. Network topology $\{x\}, \{y\}$ represent {x} minipod and {y} nodes in total, and the numbers in job configurations are the degree of DP, TP, PP. The scheduling unit is the PP group.

| Settings | Network Topology | Job Configs |
|----------|------------------|-------------|
| (i)      | 3, 18            | 12, 4, 2    |
| (ii)     | 5, 438           | 24, 4, 8    |
| (iii)    | 11, 1019         | 46, 8, 8    |

**Setups.** We use 3 settings in the benchmark as listed in Table 1, where the network topology is taken from a subset of our GPU cluster, and the job configurations are representative for small, medium, large jobs respectively. We also vary the value of $\alpha$ to investigate different degree of affinity.

Figure 7 compares the performance of different algorithms. Our algorithm consistently outperforms other baselines and up to 1.67x compared to the best baseline. On average, it leads to 1.2x reduction of the weighted sum of the maximum spread for communication groups. In the simple topology (setting *(i)*), our algorithm achieves the same score as best-fit, gpu-pack and topo-aware, because the network topology and job configurations are relatively simple, so there is no room to improve the placement. For medium and large jobs, our algorithm is better than the other baselines due to the large search space of possible placement.

We also observe that as $\alpha$ increases, our scheduling algorithm is closer to other baselines. This is because $\alpha$ controls the affinity of the DP group, and if $\alpha$ is 1, the objective function reduces to a bin-packing formulation and therefore has no difference from other bin-packing algorithms. In practice, we would not set $\alpha$ greater than 0.5 as observed from our characterization results (§4). As a result, our algorithm usually scores higher than other baselines.

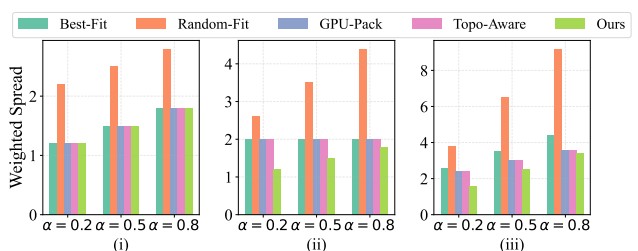

Figure 7: Weighted spread of communication groups under different scheduling algorithms.

**Scalability.** To evaluate the scalability of our algorithm, we implement an enumeration approach and compare the scheduling latency. Figure 8 investigates the scalability. The enumeration approach is guaranteed to obtain the optimal placement strategy. However, it is not scalable, as its computational complexity is $O(k^m)$ for scheduling $m$ nodes to $k$ minipods. Empirically, it can cause a scheduling

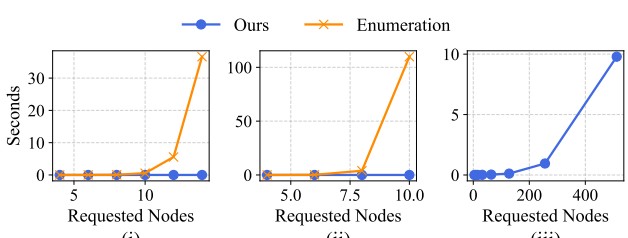

Figure 8: Scheduling latency under different configurations.

latency of 30s in a simple topology (setting *(i)*) when scheduling 14 nodes. In a medium topology (setting *(ii)*) it takes 100s+ to schedule a job with 10 nodes. In contrast, our algorithm has a low latency even if it is required to schedule a 512 node job in a cluster with 1000+ nodes.

## 7.2 Cluster Experiment

To evaluate the effectiveness of Arnold in real-world environments, we run experiments in our GPU cluster. *The specific information such as the number of GPUs and the model size, is hidden due to business concerns.* One of our LLMs is a MoE variant and was trained previously with more than 9600 GPUs (1200+ nodes). We first run the experiment by scheduling the job with 208 GPUs, and validate the speedup achieved by Arnold. We then run the pre-training at full scale. We compare Arnold with an SOTA production system for LLMs, MegaScale [18], which takes a full-stack solution to optimize LLMs training and scale to $O(10, 000)$ GPUs.

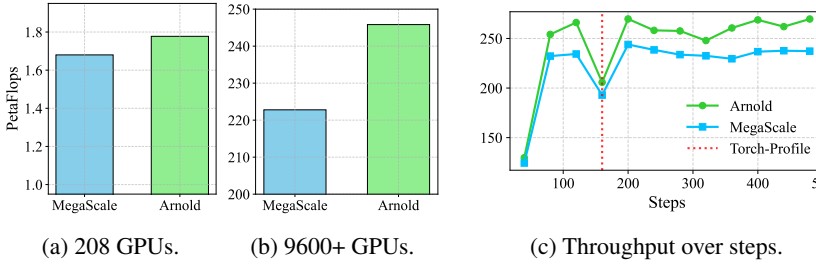

Figure 9: Cluster experiments.

**End-to-end experiments.** Figure 9a and 9b illustrate the average throughput of the two systems. Arnold achieves an average speedup of $5.7\%$ and $10.6\%$ respectively. We observe that Arnold reduces the maximum spread for the DP group and the PP group by 3x and 2x in the medium-scale experiment, while for the full-scale experiment, the reduction is 1.5x and 1.3x. This is because it is more likely to spread nodes across minipods in the cluster for medium-scale experiment if not planned carefully. However, for the full-scale experiment, the requested GPUs take up more than 50% of the total number of GPUs in the cluster, and therefore the space of scheduling is shrunk.

Nevertheless, we observe that as the model size scales and more nodes are added, the speedup achieved by Arnold also increases. The finding is consistent with Figure 5b in §4, in which the effectiveness of optimized placement is more prominent as the model size scales up. Our production

deployment encompasses significantly larger models exceeding 400 billion parameters distributed across a substantially higher number of nodes, resulting in intensive network communication demands.

Figure 9c plots the performance of the full-scale experiment over the training steps. Despite the performance fluctuation at the 160-th step due to torch profiling, we observe that the Arnold outperforms MegaScale consistently. The LPJ runs for more than one month, and the improvement is significant for downstream tasks as well as cost savings (GPU hours and human resources), since cloud providers price one Hopper GPU at $2.99/h. Moreover, the proposed optimization is orthogonal to those reviewed in §8, allowing it to be applied in conjunction with existing methods. Importantly, the optimization is fully transparent to end users.

We also evaluate the scheduling algorithms on an open-source model (Llama3 8B), replacing the InfiniBand network with a RoCE network and using 12 nodes equipped with L20 GPUs, and the results are shown in Appendix I. It indicates our scheduling algorithm generalizes well across models and hardware configurations.

**Breakdown analysis.** Figure 10 shows kernel-level examination of both systems using the Torch profiler. It reveals that communication and topology-aligned placement strategies yield a nuanced impact: while they enhance the performance of a communication kernel as expected, they simultaneously introduce performance degradation in other kernels, including a computation kernel. Through systematic ablation studies in Appendix J, we identify re-

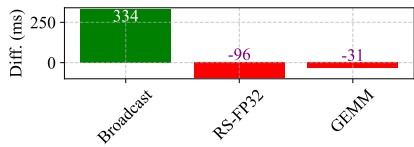

Figure 10: Breakdown analysis.

source contention between GPU streams as the fundamental mechanism underlying this phenomenon, which presents when communication and computation kernels execute concurrently across multiple streams. These observations broaden the scope of topology-aware scheduling by showing that its impact extends beyond communication efficiency, influencing the execution characteristics of computation kernels as well.

# 8 Related Works

**LLMs training.** LLMs have become a significant workload in the field of machine learning [30, 37, 49], and the computing infrastructure continues to evolve to adapt to the challenging workload. For example, efficient parallelization strategies are searched by model parallelizers [51, 22, 38], training frameworks specialized for training scalability are built to orchestrate large-scale worker nodes [18, 36, 24], and high-performance operators are developed to maximize the utilization of hardware accelerators [9, 6, 12]. However, those works are orthogonal to the optimization proposed in this paper, as the physical network topology is only visible at the cluster scheduling layer. Therefore, our work is transparent to the underlying infrastructure codes and the speedup is achieved on top of existing effort to accelerate the training performance.

**Deep learning job schedulers.** Job scheduling systems for deep learning tasks have been widely deployed by companies [7, 44, 43, 2, 25]. However, none of deep learning jobs have come even close to the scale and importance of LLM pre-training jobs. Arnold addresses this gap by providing a solution tailored to scheduling LLM pre-training jobs, complementing existing schedulers. Recent studies have begun to explore the characteristics of LLM workloads in GPU clusters [13]. In contrast, our work specifically targets the optimization of LLM pre-training performance.

**Emerging network architectures.** Emerging network architectures [40, 32] have significant impacts on future design of scheduling algorithms, and we believe that the presented algorithm is capable of handling other network topologies, for production GPU clusters follow a multi-tier design. The collective communication is bottlenecked by the highest hierarchy, i.e. the inter-minipod communication in our case. In our case, we observe little impact of intra-minipod topologies; however, we can also solve the MIP in intra-minipod level if we can observe any performance degradation.

# 9 Conclusion

In this work, we present Arnold, a scheduling system that summarizes our experience in effectively scheduling LPJs at scale. In-depth characterization is performed and a scheduling algorithm is developed to align LPJs with the topology of modern data centers. Through experiments, we show the effectiveness both in simulation-based and real-world GPU clusters.

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

## A    Limitation and Future Works

**Failure recovery.** On hardware failure, the optimal placement of LPJs will inevitably change, but it is too expensive to solve the MIP and then migrate to the new placement. A potential approach is to increase the number of GPUs of communication groups in the initial scheduling as backups, which only run preemptive jobs and can be replaced with failure nodes when needed.

**Other network topology.** The characterization results, while derived from our in-house data center environments, exhibit broad applicability to CLOS-based network topologies, which represent the predominant network architecture in modern data center deployments. By varying the affinity parameters, one can effectively trade-off the balance of DP and PP groups in their own data centers. Therefore, the characterization methodology and the scheduling algorithm are generalizable to other data centers for large-scale LPJs.

## B    Analytical Estimation for Communication Volume

To estimate the communication volume of pre-training jobs, we adopt an analytical model for GPT-based variants. We use the same notation from previous work [24] by denoting the vocabulary size $V$, global batch size $gb$, micro-batch size $mb$, sequence length $s$, hidden dimension $h$, the number of layers $l$, the DP size $dp$, the PP size $pp$, the number of VPP size $vp$, number of microbatches $m$. We have:

$$m = \frac{gb}{mb * dp} \tag{11}$$

- DP groups. GPUs within the same group replicate the model weights and exchange parameters as well as gradients, so the communication volume can be computed using Equation 12.

$$DP - volume = h * (V + s) + l/pp * (4h^2 + 2h + \underbrace{8h^2 + 7h}_{\text{dense layer}}) \tag{12}$$

  For MoE models, we can replace the number of parameters with MoE layers accordingly.

- PP groups. GPUs exchange intermediate activation to adjacent PP stages, and thus we apply Equation 13 to estimate the data volumes.

$$PP - volume = 2 * mb * s * h \tag{13}$$

## C    Sensitivity to Shared Load

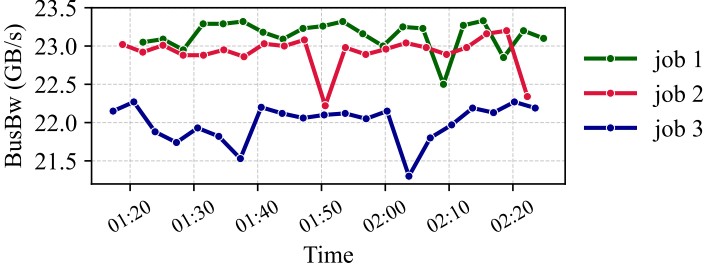

Figure 11: Bandwidth interference.

Since GPU clusters are usually multi-tenant to improve resource utilization, we also study the interference between inter-node communication quantitatively. Before we bring our cluster online, we perform large-scale stress test on our cluster by running NCCL tests. We record the time series of bus bandwidths and show in Figure 11. The stress test consists of 3 jobs, requesting thousands

of GPUs each, and spanning across multiple minipods in the cluster. Each job performs all-to-all communication with a message size of 2GB constantly, which simulates jobs performing extensive inter-node communication on a busy cluster because all-to-all generates large amount of flows in the network.

We can observe performance fluctuation for all three jobs. For example, after job 1 is launched at $01:22$, job 3 has a slight performance degradation of $0.5$GB/s ($3\%$). The maximum performance degradation is up to $5\%$ for job 3 during the stress test period. This suggests that jobs spanning a larger number of minipods not only suffer from increased bandwidth loss but are also more exposed to interference from other workloads in the cluster.

## D    Ada Lovelace GPUs

We repeat the characterization experiment in another GPU cluster, where each node is equipped with L20 GPUs, to ensure our finding is not limited to H800 GPUs, and we briefly summarize the results in Table 2.

| Best placement | Model size | speedup |
|----------------|-----------|---------|
| DP-aligned | dense 7b | 1.4% |
| PP-aligned | dense 14b | 0.5% |

Table 2: Results on L20 GPUs cluster.

We observe that DP-aligned can yield greater speedups for certain model configurations. This is likely because during training, L20 GPU uses a 8-bit data format, which halves the communication volumes between PP stages. However, DP groups still use 32-bit for parameter and gradient synchronization, so the communication volumes remain unchanged. As a result, DP group communication can become the dominant overhead, making a placement strategy that prioritizes DP groups more beneficial. However, as the size of the model grows, the bottleneck shifts back to the communication of the PP group, so the PP-aligned is preferential.

## E    Communication Matrix

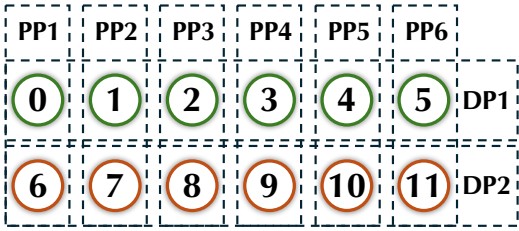

Figure 12: Example placement (96 GPUs and 12 nodes) of a LPJ.

The example placement in Figure 12 has a DP group size of 6, and PP group size of 2. Nodes in different colors represent they are placed in different minipods. The number is the rank of the node. In this example, DP group is aligned and the communication of PP group must cross spine switches.

## F    Queue Management

Algorithm 1 illustrates our queuing policy. It manages the job queue and reserve resources for the imminent LPJ. It also employs an ML-driven job completion time (JCT) predictor to balance the trade-off of queuing delay and resource utilization

**JCT predictor.** The JCT prediction enables opportunistically scheduling short-lived jobs to the reserved resources as long as they can finish before the arrival of LPJ. This helps improve resource utilization and decrease queuing delay. The prediction is based on metadata associated with jobs, such as the number of requested CPUs and GPUs, the requested amount of drives, the department of

**Algorithm 1** Job scheduling policy

---

1: $J$; //job's configurations and metadata
2: $V$; //physical view of the cluster
3: $Q$; //job queue, sorted by priority and arrival time
4: $O$; //delay list
5: **function** SCHEDULER($J, V, Q$)
6:     **while** $True$ **do**
7:         $O \leftarrow \emptyset$
8:         **while** $Q \neq \emptyset$ **do**
9:             $J \leftarrow Q.pop()$
10:             **if** $preemptable(J)$ **then**
11:                 $sched(J, V)$
12:             **else if** $J.request < V.free\_resource$ **then**
13:                 $sched(J, V)$
14:             **else if** $pred\_JCT(J) < arrival\_time$ **then**
15:                 $sched(J, V)$
16:             **else**
17:                 $O.add(J)$
18:             **end if**
19:         **end while**
20:         $Q \leftarrow O$
21:         $sleep(interval)$
22:     **end while**
23: **end function**

---

task owners, etc. Although estimating the exact JCT is inherently difficult, we adopt a coarse-grained forecasting strategy, which classifies the JCT into different time intervals.

To train the JCT predictor, we retrieve historical trace data from the database. Then, we pre-process the data, such as removing jobs that are early killed by users, and divide the JCT into 10-minute intervals. We then train models to predict the interval into which incoming jobs may fall by the metadata associated with the jobs. We tried both a deep neural network (DNN) and a gradient boosting predictor (GBM)[20], and found that GBM achieves higher performance, likely due to its ability to handle categorical variables.

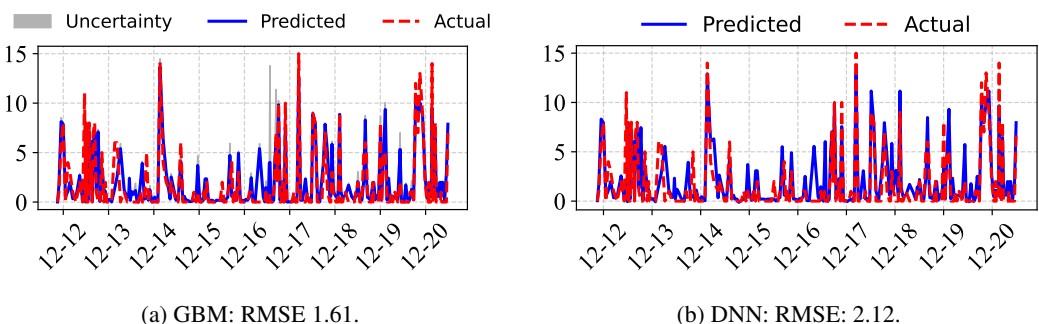

(a) GBM: RMSE 1.61.          (b) DNN: RMSE: 2.12.

Figure 13: JCT prediction.

To demonstrate effectiveness, we extract 4-month trace data and divide them into a training set ($90\%$) and a test set ($10\%$). Figure 13 shows an example of prediction in the test set. We apply randomized grid search to optimize hyper-parameters and also use bagging to determine uncertainty estimation. We observe that the RMSE is $1.61$ in the test set, and recent studies suggest that such prediction could help scheduling decision [42, 3].

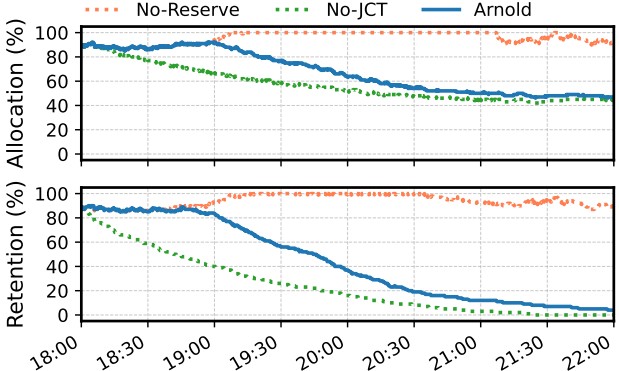

Figure 14: Allocation and retention rate over time since a LLM pre-training job is planned.

## G  Evaluation of Queue Management

We collect job traces and replay in the trace-driven simulator. Figure 14 shows the allocation and retention rate over time. The allocation rate is determined by dividing the total number of nodes by the nodes with allocated jobs. The retention rate measures how many planned nodes for the LLMs job are occupied by other jobs, which inevitably requires manual preemption when the LLMs job arrives. At $18:00$, Arnold is told the arrival time of the LLMs job at $22:00$, and so it triggers the code path to plan and reserve resources accordingly. We use a default bin-pack algorithm to schedule other jobs.

The allocation rate is $0.9$ initially and then gradually decreases to below $0.5$ due to the need to reserve 1200+ nodes in the cluster. At the beginning of the imminent period, the retention rate is relatively high and matches the allocation rate because the previous allocated jobs are not aware of the incoming LLMs pre-training job. Then the retention rate decreases faster than the allocation rate since nodes have been reserved, and is close to $0$ at the end of the period, showing the effectiveness of the scheduling policy.

For the reserving-and-packing policy [42], it does not offer strong semantics for reservation (i.e. best effort). Thus, the scheduler will not be able to generate a feasible solution as the LLM job arrives, not to mention optimal placement (orange line). The JCT prediction navigates the trade-off space between resource utilization and guarantees by scheduling opportunistically short-lived jobs to the reserved zone. In its absence, both queuing delays and resource idle times increase, as indicated by the green line.

## H  Example MIP solution

Table 3: The MIP solution values for setting (i) in §7.1.

| Index | 0 | 1 | 2 |
|-------|-----|-----|-----|
| s_0_* | 0.0 | 0.0 | 1.0 |
| s_1_* | 0.0 | 0.0 | 1.0 |
| s_2_* | 0.0 | 0.0 | 1.0 |
| s_3_* | 0.0 | 1.0 | 0.0 |
| s_4_* | 0.0 | 1.0 | 0.0 |
| s_5_* | 0.0 | 1.0 | 0.0 |

| Index | 0 | 1 | 2 |
|-------|-----|-----|-----|
| p_0_* | 0.0 | 0.0 | 1.0 |
| p_1_* | 0.0 | 0.0 | 1.0 |
| p_2_* | 0.0 | 0.0 | 1.0 |
| p_3_* | 0.0 | 1.0 | 0.0 |
| p_4_* | 0.0 | 1.0 | 0.0 |
| p_5_* | 0.0 | 1.0 | 0.0 |

| Index | Value |
|-------|-------|
| y[0]  | 0.0   |
| y[1]  | 1.0   |
| y[2]  | 1.0   |

# I  Evaluation on open-source models

Table 4: Average throughput (PetaFlops) comparison on Llama3 8B.

| Scheduling algorithm | llama3 8B |
|----------------------|-----------|
| Bin-pack (used by MegaScale) | 3.80 |
| GPU-pack | 3.82 |
| Ours | 3.86 |

## J Break-down analysis

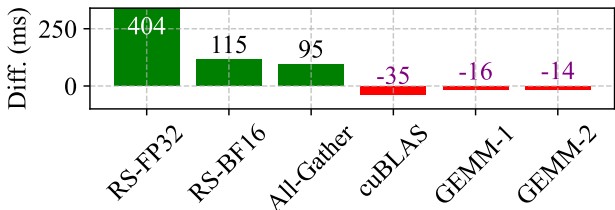

Figure 15: Breakdown analysis of the ablation experiment.

We compare aggregated kernel-level metrics by summing the duration for each type of kernel. Figure 10 illustrates the speedup achieved by Arnold (in green) and the slowdown (in red) of the full-scale experiment. We only report kernels whose difference is significant. The most significant speedup is the broadcast kernel ($10\%$), which is the optimized P2P implementation of our communication library. However, the speedup is slightly offset by the slowdown of a reduce-scatter kernel and even a computational kernel. The slowdown contradicts our expectations, as Arnold's scheduling also reduces the spread of DP groups. Moreover, we have not changed other configurations, so the slowdown of the GEMM kernel is unexpected.

After thorough investigation, we suspect the slowdown is due to the interference between GPUs' streams. Due to hybrid parallelism, GPUs maintain multiple streams that issue operations concurrently during training. Although overlapping computation with communication indicates good performance optimization, it also causes resource contention and interference.

**Network topology affects computation kernels.** To investigate the counter-intuitive results, we isolate the impact of streams by modifying NCCL. For example, we add additional environmental variables such as *NCCL_DP_MIN_NCHANNELS* to have fine-grained controls on the DP stream. We disable channel auto-tuning and rerun jobs with and without setting the NCCL variable. Figure 15 shows the breakdown analysis. Communication kernels have speedups by setting the NCCL variable, whereas computation kernels have slowdown. Since the only change is the NCCL variable, it indicates if we allocate more GPU SMs to communication, computation kernels suffer from performance loss for less available SMs.

In production training, the NCCL variables are dynamically auto-tuned, so given that network topology-optimized scheduling influences the communication of DP and PP groups, ultimately it causes variations in the computation kernels.

