# OpenReview forum: "Efficient Pre-Training of LLMs via Topology-Aware Communication Alignment on More Than 9600 GPUs"
_NeurIPS.cc/2025/Conference — NeurIPS 2025 poster_

### Official Review · Reviewer_ShkR · 2025-06-02

**Clarity:** 3
**Significance:** 3
**Originality:** 4
**Rating:** 5
**Confidence:** 4

**Summary:**

This paper introduces Arnold, a topology-aware scheduling system designed to optimize the placement of large language model (LLM) pre-training jobs (LPJs) in modern GPU clusters. The authors argue that existing schedulers fail to consider the complex, high-volume but sparse communication patterns typical in LLM training, leading to inefficient resource usage and prolonged training times.

**Questions:**

Here are additional comments for weakness 1:

1. After reading twice, I can basically understand that the DP and PP make topology-related problems in this paper different from earlier works like https://www.comp.nus.edu.sg/~hebs/course/cs6284/papers/zhang-sigmod19.pdf ,BriskStream: Scaling Data Stream Processing
on Shared-Memory Multicore Architectures. SIGMOD 2019

2. But the understanding is fully formula-based and fragmented to me; it could be better if the authors use a concrete paragraph to explain intuitions and summarise the differences.

**Ethical Concerns:**

["NO or VERY MINOR ethics concerns only"]

**Final Justification:**

I am satisfied with response and will keep my score

**Limitations:**

N.A.

**Paper Formatting Concerns:**

N.A.

**Quality:**

4

**Strengths And Weaknesses:**

Strengths:

1. Demonstrates clear empirical benefits with both simulation and production-scale experiments.

2. Introduces a principled scheduling model that adapts to diverse LLM configurations and network conditions.

3. Well-integrated system with resource management and job queue handling, including ML-based job completion time prediction.

Weakness:

1. Presentation issues: the solution is novel after a careful examination of the formulation, but novelty is not discussed well at a glance.

2. Lack of discussion about data compression and emerging networking architectures, which are also impactful in distributed training.

---

> ### Author Rebuttal · Authors · 2025-07-30
>
> > **w1 + q1 + q2: Presentation issues. But the understanding is fully formula-based and fragmented to me; it could be better if the authors use a concrete paragraph to explain intuitions and summarise the differences.**
>
> We apologize for the presentation issues, and will clarify in the text.
>
> Multi-core CPU parallelism (as in BriskStream) and inter-GPU parallelism (as in LLM training) both aim to use multiple processing units to speed up computation. However, they are fundamentally different due to their underlying memory architecture, communication mechanisms, and architectural granularity.
>
> **Memory architecture:** Multi-core Parallelism is primarily concerned with efficiently managing a shared memory space among cores, with the main challenge being the performance penalty of Non-Uniform Memory Access (NUMA).
>
> Inter-GPU Parallelism is primarily concerned with orchestrating computation across multiple devices that have their own separate and distributed memory, with the main challenge being the high cost of explicitly transferring data between them.
>
> **Communication mechanisms:** Multi-core Parallelism happens implicitly through memory. When a producer operator writes a tuple to a queue and a consumer operator reads it, they are accessing a shared location in RAM.
>
> Inter-GPU Parallelism is explicitly managed via libraries like NCCL (NVIDIA Collective Communications Library).
>
> **Architectural granularity:** Multi-core Parallelism is designed for task parallelism on a single tensor computation level. For example, matrix multiplication for tensors operations.
>
> Inter-GPU Parallelism is designed for massive data parallelism on homogeneous, throughput-oriented, and highly regular workloads. For example, a group of communication processes.
>
> > **w2: Lack of discussion about data compression and emerging networking architectures, which are also impactful in distributed training.**
>
> We will add more discussion on data compression pipeline and networking architectures. Reviewer is right about these factors can impact distributed training.
>
> **Data compression and processing.** Data processing typically takes place on CPU and causes GPU idle. We adopt MegaScale's optimization, by overlapping data processing with gradient synchronization. They can be fully overlapped because they take place in different devices and the data processing overhead can be hidden.
>
> **Emerging networking architectures.** Emerging network architectures have significant impacts on the future design of scheduling algorithms. We think the presented algorithm is in principle able to handle other netowrk topology.
>
> We observe that production GPU clusters follow the multi-tier design (Alibaba’s HPN, NVIDIA’s DGX). The collective communication is bottlenecked by the highest hierarchy - the inter-minipod communication in our case. In our case, we observe little impact of intra-minipod topology, however, we can also solve the MIP in intra-minipod level if we can observe any performance degradation.
>
> There are more dedicated network architectures exist for LLM training, e.g. rail-only. However, the inability of any-to-any connectivity make it unlikely to be adopted in production training clusters in the near future.

---

> > ### Comment · Reviewer_ShkR · 2025-08-02
> >
> > Thanks for the response. I am satisfied with the clarification.

---

> > > ### Author Response · Authors · 2025-08-04
> > >
> > > We thank the reviewer for the acknowledgement. We are happy to provide more information if needed.

---

### Official Review · Reviewer_e9bD · 2025-06-30

**Clarity:** 3
**Significance:** 4
**Originality:** 3
**Rating:** 5
**Confidence:** 4

**Summary:**

LLM training spends a significant amount of time on collective communication. However, existing infrastructures are not optimized for this type of communication pattern, where GPUs mostly communicate within groups. The key challenge is that data parallelism (DP) and pipeline parallelism (PP) are orthogonal, making it hard to arrange workloads such that all traffic avoids the spine network. This paper proposes **Arnold**, a workload-aware training framework that uses Integer Linear Programming (ILP) to search for optimal job placement that minimizes inter-minipod communication, achieving real-world speedups.

**Questions:**

Thanks for submitting to NeurIPS. The paper motivates the problem of misaligned GPU communication requirements with the underlying hardware, which is both educational and practically important. I enjoyed reading the paper. Overall, I think it addresses a timely and significant challenge. I have a few clarifications I’d like to ask:

1. **Figure 4 Analysis**
   Figure 4 shows the profiling results across different numbers of minipods. After 4 minipods, the performance degrades drastically. Why is this the case? Is it due to the physical connectivity between minipods?

2. **Bandwidth vs. Latency Bottlenecks**
   Is the performance degradation primarily caused by limited bandwidth between minipods, or by increased latency due to additional switches in the communication path?

3. **Scaling Model Size**
   The paper shows that adding more layers to increasing model size makes Arnold more beneficial. Could you clarify why this is the case? Although the network becomes more important, compute cost also increases with model size.

4. **ILP Approximation Details**
   The ILP formulation in equations (2) and (3) is clear. However, I am struggling to understand the approximation version in equations (4)–(10). Could you elaborate more on the rationale and implementation behind these approximations?

5. **Network Topology Co-design**
   The paper focuses on scheduling-side solutions. If you had the ability to redesign the data center network topology, what strategies would you suggest to better accommodate LLM training workloads?

**Ethical Concerns:**

["NO or VERY MINOR ethics concerns only"]

**Final Justification:**

The idea of the paper is novel and important. While the clarity can be improved, the rebuttal response is satisfactory. I would like to keep the score.

**Limitations:**

Yes

**Paper Formatting Concerns:**

No concerns

**Quality:**

4

**Strengths And Weaknesses:**

## Strengths
- The problem is well-motivated and critical for large-scale LLM training.
- Evaluation is based on real-world training runs, which is a strong validation of the system.
- The proposed method effectively balances tensor parallelism (TP) and data parallelism (DP).
- Arnold is shown to scale to clusters with thousands of GPUs.

## Weaknesses
- The description of the ILP formulation is unclear in parts.
- No example of the search output is presented, which would help readers better understand the solution.

---

> ### Author Rebuttal · Authors · 2025-07-30
>
> > **w1: The description of the ILP formulation is unclear in parts.**
>
> We apologize for the presentation and will add more clarification in the text.
>
> In Equation (2) each node is a scheduling unit, while in Equation (4) the communication group (e.g. PP) becomes a scheduling unit.
>
> **A simple example.** Suppose we have $m$ DP group and $n$ PP group, after the coarsening, the scheduling problem becomes “how to put the $n$ unit into $k$ minipods?” - we are allowed to spread one unit across multiple minipods, but we want to minimize the spread, hence introducing the $T$ auxiliary variable. On the other hand, DP groups’ spread is “orthogonal” to PP group – GPUs in the same stage of different PP groups belong to the same DP group, so if each unit (PP) is packed into a different minipod (the optimal $T$), the spread of DP groups is maximized. Thus, we also minimize the overall usage of minipod, which controls the spread of DP groups.
>
> Combining the above, Equation (4) is our optimizing objective function.
>
> Equation (5) constraints the maximum spread of PP groups less or equal to $T$, and $T$ is being minimized.
>
> Equation (6) says that the allocation cannot exceed the capacity of a minipod, e.g. if a minipod has $10$ nodes free, we cannot allocate more than that. The capacity is normalized by the number of node in a PP group.
>
> Equation (7) says that while we can spread a PP group across minipods, the summation of the allocation percentage must be equal to $1$.
>
> Equation (8) says that if we allocate to a minipod, that it is considered used in the solution, i.e. $y_j = 1$.
>
> Finally, Equation (9) and (10) define the range of the binary variable $y_j$ and $s_{ij}$, and continuous variable $p_{ij}$.
>
> ---
>
> > **w2: No example of the search output is presented, which would help readers better understand the solution.**
>
>
> With the explanation in **w1**, the search output is the specific values of each variable, i.e. $y_j$, $s_{ij}$ and $p_{ij}$. For example, $y_j$ indicates whether the $j$-th minipod is used. $p_{0,0}$ tells us how many nodes of the $0$-th PP group should be allocated to the $0$-th minipod. Thus, we use the value of $p_{i,j}$ to know which PP group goes to which minipods.
>
>
> ---
>
> > **q1: Figure 4 Analysis**
>
> The degradation is due to the increased routing distance and the switch is overwhelmed.
>
> The bandwidth degradation is expected as communication spans more minipods due to the increased physical routing distance. The specific 4-5 minipod degradation may be due to the switch is overwhelmed by the hundreds of bursty flows generated by the communication group over a short period of time, so the ECMP load balancing algorithm is not effective. The flows collide to certain links and cause link saturation, and the saturated link becomes the straggler to slow down the entire communication process.
>
> ---
>
> > **q2: Bandwidth vs. Latency Bottlenecks**
>
> As discussed in the response to **q1**, the increased physical routing distance and the collision of network flows lead to performance degradation. This is reflected by the measured bandwidth. The theoretical bandwidth is the same for inter-minipods connection.
>
>
> ---
>
>
> > **q3: Scaling Model Size**
>
> The training is bottlenecked by communication, and the optimized topology benefits GPU utilizes resources for communication, which is more pronounced when both computation and communication sizes increase.
>
> Computation is much faster than communication for GPUs (TeraFlops vs GB/s). Thus, computation kernels are blocked by the dependent communication kernels. In Figure 10, we observe that an optimized topology benefits communication, which effectively optimizes the bottleneck of training.
>
>
> ---
>
> > **q4: ILP Approximation Details**
>
> As explained in response to **w1**, we have clarified the details of each equations:
>
> Equation (4) is the main objective function that minimizes the spread of PP groups and DP groups.
>
> Equation (5) constraints the maximum spread of PP groups less or equal to $T$, and $T$ is being minimized.
>
> Equation (6) says that the allocation cannot exceed the capacity of a minipod, e.g. if a minipod has $10$ nodes free, we cannot allocate more than that. The capacity is normalized by the number of node in a PP group.
>
> Equation (7) says that while we can spread a PP group across minipods, the summation of the allocation percentage must be equal to $1$.
>
> Equation (8) says that if we allocate to a minipod, that it is considered used in the solution, i.e. $y_j = 1$.
>
> Finally, Equation (9) and (10) define the range of the binary variable $y_j$ and $s_{ij}$, and continuous variable $p_{ij}$.
>
> ---
>
> > **q5: Network Topology Co-design**
>
> We think the presented algorithm is in principle able to handle other netowrk topology.
>
> We observe that production GPU clusters follow the multi-tier design (Alibaba’s HPN, NVIDIA’s DGX). The collective communication is bottlenecked by the highest hierarchy - the inter-minipod communication in our case. In our case, we observe little impact of intra-minipod topology, however, we can also solve the MIP at the intra-minipod level if we can observe any performance degradation. In short, at every level of the network hierarchy, we can solve the MIP to find the optimal placement at that level.

---

> > ### Comment · Reviewer_e9bD · 2025-08-07
> >
> > Thank you for your clarifications. The response addresses most of my comments.
> >
> > However, I remain unsure about the claim that *"computation is much faster than communication for GPUs"*. For example, MegaScale demonstrates 55% MFU utilization, which suggests that compute kernels may still be a significant contributor to overall training time.
> >
> > Additionally, it would be helpful if the authors could include concrete values for $y_j$, $s_{ij}$, and $p_{ij}$ in the paper, as this would provide readers with a clearer understanding of the optimized topology.

---

> > > ### Author Response · Authors · 2025-08-07
> > > **response**
> > >
> > > > However, I remain unsure about the claim that "computation is much faster than communication for GPUs". For example, MegaScale demonstrates 55% MFU utilization, which suggests that compute kernels may still be a significant contributor to overall training time.
> > >
> > > Compute kernels are significant contributors. Ideally, GPUs should spend 100\% of the time in computation, which pushes the throughput to the device peak. Communication is considered as "overhead", because it is not doing useful work. In short, training should be as compute-bound as possible.
> > >
> > > At large scales, communication efficiency reduces due to reasons such as greater inter-node distances, flow collision and network congestion from increased data volumes. The training is shifting towards communication-bound, so the impact of optimized topology placement on overall performance is magnified.
> > >
> > > > Additionally, it would be helpful if the authors could include concrete values for $y_j$, $s_{ij}$, and $p_{ij}$ in the paper, as this would provide readers with a clearer understanding of the optimized topology.
> > >
> > > Thanks for pointing out the issue. We will include the concrete values in the Appendix for experiments in section 7.1. For example, the output of the scheduling in setting (i) is shown below:
> > >
> > > | Index    | 0 | 1 | 2 |
> > > |----------|-----|-----|-----|
> > > | s_0_*    | 0.0 | 0.0 | 1.0 |
> > > | s_1_*    | 0.0 | 0.0 | 1.0 |
> > > | s_2_*    | 0.0 | 0.0 | 1.0 |
> > > | s_3_*    | 0.0 | 1.0 | 0.0 |
> > > | s_4_*    | 0.0 | 1.0 | 0.0 |
> > > | s_5_*    | 0.0 | 1.0 | 0.0 |
> > >
> > > | Index    | 0 | 1 | 2 |
> > > |----------|-----|-----|-----|
> > > | p_0_*    | 0.0 | 0.0 | 1.0 |
> > > | p_1_*    | 0.0 | 0.0 | 1.0 |
> > > | p_2_*    | 0.0 | 0.0 | 1.0 |
> > > | p_3_*    | 0.0 | 1.0 | 0.0 |
> > > | p_4_*    | 0.0 | 1.0 | 0.0 |
> > > | p_5_*    | 0.0 | 1.0 | 0.0 |
> > >
> > > | Index | Value |
> > > |-------|-------|
> > > | y[0]  | 0.0   |
> > > | y[1]  | 1.0   |
> > > | y[2]  | 1.0   |

---

### Official Review · Reviewer_awuk · 2025-06-30

**Clarity:** 3
**Significance:** 2
**Originality:** 2
**Rating:** 3
**Confidence:** 4

**Summary:**

This paper introduces a system (Arnold) that improves LLM pre-training through a scheduling algorithm that takes into account data center topology to be more communication-aware. Claimed contributions are investigations into existing data center training, introduction of the design of Arnold, as well a resource scheduling policy, and evaluations of this system. The paper goes through various motivating experiments before introducing the scheduling algorithm.

**Questions:**

### Major

Why isn’t equation (4) symmetric? $\alpha$ and $\beta$ are symmetric, but T is the max spread across PP communicator groups as I understand, but for DP communicator groups all the minipods are summed. Wouldn’t it make more sense to have a max spread measure for this as well?

Why isn't $T$ penalized non-linearly?

Are there any experiments against an external baseline (MegaScale) with experimental settings that can be disclosed?

### Minor

Is there any explanation for why all gather and send/recv bandwidths degrade over multiple minipods? Is this due to NCCL using a different algorithm? Perhaps the algorithm isn’t being selected intelligently going from 4 to 5 minipods.

Lines 175-177 discuss how training is insensitive to intra-minipod topology. But Figure 3 shows an intra-minipod example. Why is this?

**Ethical Concerns:**

["NO or VERY MINOR ethics concerns only"]

**Final Justification:**

Overall, this paper is strong. I am still on the fence about it.

I reiterate my statement from the Strengths and Weaknesses section:

> Overall, this paper is a welcome study in pre-training efficiency, and I am looking forward to the open sourced code. Clarity is good, with sufficient background introduced. I am concerned about whether the theoretical formulation of the MIP makes sense (and hope this will be clarified in the rebuttal) but more importantly, whether there is enough significance/novelty as a research work suitable for NeurIPS - the main research contributions are elucidating the DP/PP misalignment and introducing the MIP although there is clearly a significant amount of engineering that went into the work.

The authors have satisfied me with their explanation of the MIP.

Strengths:

- Great speedup
- Well written
- Good motivational examples
- Theoretical formulation makes sense
- Timely topic
- Great fit for NeurIPS

Cons:

- Not sure there is enough novel material and insight.
- Still not very satisfied with the E2E examples even with Llama 8B. Would prefer more examples.

**Limitations:**

Yes.

**Paper Formatting Concerns:**

None.

**Quality:**

2

**Strengths And Weaknesses:**

I would first like to thank the authors for submitting their work to NeurIPS 2025.

### Pros

This paper is interesting as it identifies two key problems with current cluster schedulers, namely the misalignment of schedulers with data center topology, and PP/DP placement trade off.

The background is overall clear, and has a good exposition for DP/PP/TP. It also includes easy to understand figures for Hybrid parallelism and an example of a data center topology.

Section 3 and 4 also add additional exposition for the reader, and set up the problem in a very nice way. I like the background given on both the data center topology as well as the motivating experiments. Particularly, the communication operation performance and E2E training performance motivating examples are useful for understanding. Additionally, the E2E training performance motivating example is taken over a few different settings including MoE and Dense models.

The system design/architecture makes sense, and seems like it would work in practice.

### Cons

In line 82 (DP), it can be made more clear that this is different from pure DP and is in fact FSDP. It would also be good to at least spend a couple of lines on some form of context/sequence parallelism to show that that’s been acknowledged even though that will not make a difference from the DP/PP perspective.

In figure 4(a), it is not clear where the bandwidth is measured between, i.e. is this within a minipod.

I think the biggest issue with sections 3 and 4 is that it would be good to show what the alignment pattern is with SoTA communication frameworks - i.e. demonstrate that this is not actually what occurs in practice today, as it seems it would be fairly simple to align by PP or DP.

I may have misunderstood, but it seems that section 4 implies that $T$ should be penalized in a non-linear manner as collective performance varies with the number of minipods in a nonlinear manner. This is not reflected in the MIP.

Although the distance to nearest database entry in section 5.2 makes sense, this seems like it will be inaccurate for models with a large euclidean distance with the current entries. In other words, this won’t generalize easily to other large training jobs (for example, different size, parameter shape, GPU) without running a characterization experiment.

Another issue here is the lack of discussion regarding related work. I see this is in Appendix A, but should be part of the main paper, especially since the experimental results are only against 1 external baseline. In addition, it is hard to make sense of the results when the specific number of GPUs and model size is hidden. In my humble opinion, a much shorter job, but with an open source model and arbitrary number of GPUs would make much more sense as an experiment from a reproducibility and use point of view.

---

Overall, this paper is a welcome study in pre-training efficiency, and I am looking forward to the open sourced code. Clarity is good, with sufficient background introduced. I am concerned about whether the theoretical formulation of the MIP makes sense (and hope this will be clarified in the rebuttal) but more importantly, whether there is enough significance/novelty as a research work suitable for NeurIPS - the main research contributions are elucidating the DP/PP misalignment and introducing the MIP although there is clearly a significant amount of engineering that went into the work.

#### Other Comments

Nits:

Line 109 - include Infiniband bandwidth if possible.

Figure 3(a) - Suggest to call this “Fully misaligned”

Section 3 - Add a forward pointer to section 4 E2E performance.

Figure 2(b) can also be made clearer - what is S-SW, L-SW, H, etc? Readers familiar with data center topology will know about spine-leaf topology. Should be stated in the caption or text. Also, In lines 100-109 the notation s0 is used for leaf, and s1 for spine, but this is nowhere in the figures.

Line 199 - assignment -> assignment(s)

Equation (3). Notation is a little confusing.

Line 265 - JCT should be full-form the first time. JCT (Job completion time).

---

> ### Author Rebuttal · Authors · 2025-07-30
>
> > **q1: Why isn’t equation (4) symmetric? $\alpha$ and $\beta$ are symmetric, but $T$ is the max spread across PP communicator groups as I understand, but for DP communicator groups all the minipods are summed. Wouldn’t it make more sense to have a max spread measure for this as well?**
>
> **Asymmetry of Equation (4).** Equation (2) is symmetric because each node is a scheduling unit. Equation (4) is asymmetric because we coarsen the communication group (e.g. PP) as a scheduling unit. After the coarsening, we lose the granularity to express the spread of DP groups directly, but we can still control that by minimizing the usage of minipods.
>
> **A simple example.** Suppose we have $m$ DP groups and $n$ PP groups, after the coarsening, the scheduling problem becomes “how to put the $n$ unit into $k$ minipods?” - we are allowed to spread one unit across multiple minipods, but we want to minimize the spread, hence introducing the $T$ auxiliary variable. On the other hand, DP groups’ spread is “orthogonal” to PP group – GPUs in the same stage of different PP groups belong to the same DP group, so if each unit (PP) is packed into a different minipod (the optimal $T$), the spread of DP groups is maximized. Thus, we also minimize the overall usage of minipod, which controls the spread of DP groups. Equation (4) combines the two objectives.
>
> ---
>
> > **q2: Why isn't $T$ penalized non-linearly?**
>
> We acknowledge $T$ could be penalized non-linearly because in Figure 4, it has non-linear bandwidth degradation. We have also thought about constraint $T<=4$. However, it is not feasible in production scenario.
>
> We choose the current solution due to multi-tenancy. A running cluster has dynamic job submission, hardware failure and bursty network flows. A hard constraint $T<=4$ may fail to solve at real time and it is unacceptable. A soft constraint requires quantification of bandwidth degradation, which is often interfered by other jobs, as shown in Figure 11. Besides, soft constraints are hard to express for MIP solvers.
>
> Another issue is the resource contention in Figure 10 – computation and communication kernels are overlapped for maximized efficiency during training, but they also interfere with each other, which exacerbates the difficulty of quantifying the bandwidth decay.
>
> ---
>
> > **q3: Are there any experiments against an external baseline (MegaScale) with experimental settings that can be disclosed?**
>
>
>
> Yes. We disclose the following information:
>
> | Model Size | Heads | Hidden Size | Layers | Experts  |
> |------------|-------|-------------|--------|----------|
> | 30B(medium)| 32   | 4096        | 8     | 128      |
> | 440B(large)| 32   | 6144        | 80     | 128      |
>
> The experiment settings at framework level are the same, i.e.,
>
> - overlapping DP, TP and PP.
> - developping high-performance operators.
> - applying asynchronous data pipeline.
> - tuning the network performance.
>
> The difference is the scheduling algorithm, and it is the goal of this paper to present a scheduling solution for LLM pre-training. Framework-level optimization is orthogonal.
>
> ---
>
> > **q4: Is there any explanation for why all gather and send/recv bandwidths degrade over multiple minipods? Is this due to NCCL using a different algorithm? Perhaps the algorithm isn’t being selected intelligently going from 4 to 5 minipods.**
>
> **Degradation reasons.** The degradation is due to the increased routing distance and the switch is overwhelmed. The bandwidth degradation is expected as communication spans more minipods due to the increased physical routing distance. The specific 4-5 minipod degradation may be due to the switch is overwhelmed by the hundreds of bursty flows generated by the communication group over a short period of time, so the ECMP load balancing algorithm is not effective. The flows collide to certain links and cause link saturation, and the saturated link becomes the straggler to slow down the entire communication process. This means the scheduling algorithm needs careful design, not just relying on the network tuning.
>
> **NCCL algorithms and network tuning.** NCCL uses ring algorithm for the highest bandwdith. The algorithm is tuned by NCCL's cost model. Our network software stack has been tuned meticulously, as observed from the S-curves over the message sizes in Figure 4 (a), which is detailed in NVIDIA’s NCCL performance tuning technical blog.
>
> ---
>
> > **q5: Lines 175-177 discuss how training is insensitive to intra-minipod topology. But Figure 3 shows an intra-minipod example. Why is this?**
>
> Figure 3 is an illustrative example of the alignment of communication and topology. The network architecture is hierarchical, so the illustrative example can be naturally extended to higher layer in the hierarchy. We will refine the figure to be clear about the network topology.
>
> ---
>
> > **con1: In line 82 (DP), it can be made more clear that this is different from pure DP and is in fact FSDP**
>
> It is DP, because FSDP is zero $3$ strategy that shards weights, gradients and optimizer states across all processes in DP groups. It introduces large communication overhead during forward pass. Our DP implementation is zero $1$.
>
> ---
>
> > **con2: In figure 4(a), it is not clear where the bandwidth is measured between, i.e. is this within a minipod.**
>
> This is measured between minipods. We will clarify this in the text.
>
> ---
>
> > **con3: I think the biggest issue with sections 3 and 4 is that it would be good to show what the alignment pattern is with SoTA communication frameworks**
>
> Existing SoTA frameworks use the bin-pack strategy, and it has poor alignment as shown in Figure 3. The misalignment issue of a small training job is less significant due to the small data volume. In Figure $5 (b)$, we empirically observe smaller jobs suffer less from misaligned performance degradation.
>
> However, due to the scale of LLM training, the communication volume increases and the communication patterns become very sparse, as discussed in Figure 1. Thus, the misalignment problem is much more severe, and LLM pre-training needs a more specific scheduling algorithm.
>
> ---
>
> > **con4: this won’t generalize easily to other large training jobs (for example, different size, parameter shape, GPU) without running a characterization experiment.**
>
> Our scheduling algorithm is LLM-specific and LLMs have unified model architecture (decoder-only architecture), increased model sizes by stacking more layers (7B, 33B, 70B), and hybrid parallelism. This makes it easy to pre-characterized automatically and generate database entries, i.e. launch cron-jobs and record the throughput.
>
> ---
>
> > **con5: Another issue here is the lack of discussion regarding related work. I see this is in Appendix A, but should be part of the main paper, especially since the experimental results are only against 1 external baseline. In addition, it is hard to make sense of the results when the specific number of GPUs and model size is hidden.**
>
> We will move the discussion of related work to main body in the final version with an extra page. We will add more discussion of exising DL scheduling, and why they are not sufficient for LLM pre-training jobs - mainly due to ignoring the sparse communication patterns as shown in Figure 1.
>
> Additionally to the response to **q3**, where we release details model configurations and framework-level experiment settings. The largest job uses $1216$ nodes and each node has $8$ GPUs.
>
> ---
>
> > **con6: I am concerned about whether the theoretical formulation of the MIP makes sense (and hope this will be clarified in the rebuttal) but more importantly, whether there is enough significance/novelty as a research work suitable for NeurIPS**
>
> The theoretical formualtion is clarified in the response to **q1**.
>
> **Suitability.** The topic of "Infrastructure (e.g., libraries, improved implementation and scalability, distributed solutions)" is explicitly included in the call for paper statement in NeurIPS 2025. The scope of our paper falls in the infrastructure track, which includes improved implementation, scalability and distributed solutions.
>
> **Significance/novelty.** LLMs' learning is benefited by large-scale training, which necessitates innovative infrastructure support. In section 3 and 4, we share our insights and lessons learned from large-scale training on real-world clusters. In section 5, 6 and 7, we present a scheduling algorithm capable of handling very large-scale training jobs. We hope our work shed light on scalable training solutions and benefit the community.
>
> ---
>
> > **Other comment:**
>
> We thank the reviewer for pointing out the grammar issues and lack of explanation in the caption. We will fix the grammar and add more text in the captions to make the terminology clear.

---

> > ### Comment · Reviewer_awuk · 2025-08-05
> > **Acknowledgment and Response**
> >
> > Thanks to the authors for providing this rebuttal. Overall, this clears up a lot of the questions I had.
> >
> > Outstanding issues:
> >
> > 1. I still believe the experimental results overall can be improved.
> >     - What about experiments to justify lines 247 - 250?
> >     - Any other baselines?
> >     - It is still important to provide some idea of the experimental setup for reproducibility purposes. I appreciate that there are business concerns, but then some proxy setup could be used to obtain more experiments.
> >     - What about different types of models (different MoE config, or non-MoE), or data center network? Some small result could have been used here.
> >
> > 2. Suitability of work
> >     - To be clear, I have no issue at all with the suitability of the work *from the point of view of topic*. The topic is indeed a fit for the Infrastructure track and the claimed results (10%) is also a great result for such an important/well optimized workload.
> > My issue is with the significance/novelty of the *research contributions* which are mainly just the formulation of the MIP and observation regarding DP/PP misalignment. The work though has a significant amount of engineering contribution.
> >
> > I will tentatively keep my score the same pending author's responses.

---

> ### Author Response · Authors · 2025-08-06
> **Response**
>
> > - What about experiments to justify lines 247 - 250?
>
> Lines 247-250 say that we use proportionally small-scaled models and small numbers of GPUs to approximate the performance of large-scale experiments. This is well-studied in Megatron-LM[1], in section 5.1.1, termed **weak scaling**. A more up-to-date result is also shown in Megatron-LM's official Github, under the section of "Training Speed and Scalability", where it is clear that throughput linearly increases as both model sizes and the number of GPUs scale up, and therefore the performance characteristics are comparable.
>
> [1] Mohammad Shoeybi, Mostofa Patwary, Raul Puri, Patrick LeGresley, Jared Casper, and Bryan Catanzaro. Megatron-lm: Training multi-billion parameter language models using model parallelism, 2020.
>
> > - Any other baselines?
>
> We add the GPU-packing scheduling algorithm as a baseline and integrate it into the scheduling system, so that we can compare its performance with Arnold in the Llama3 experiment shown later.
>
> > - It is still important to provide some idea of the experimental setup for reproducibility purposes. I appreciate that there are business concerns, but then some proxy setup could be used to obtain more experiments.
>
> For the training framework, MegaScale is open source, and it is capable of being used to train large models. For the scheduling layer, we build on top of Kubernetes, which is also open source. By utilizing those open-source systems, one can reproduce the experimental results.
>
>
>
> > - What about different types of models (different MoE config, or non-MoE), or data center network? Some small result could have been used here.
>
> We add an additional experiment on Llama3 8B, replacing the InfiniBand network with a RoCE network and using 12 nodes equipped with L20 GPUs. Average throughput (PetaFLOPs):
>
> | Scheduling algorithm   | llama3 8B  |
> |--------------          |---------   |
> | Bin-pack (used by MegaScale)  | 3.80|
> | GPU-pack                      | 3.82|
> | Ours                          | 3.86|
>
> The result is consistent with our findings - our scheduling algorithm achieves better speedups, but the speedup is more prominent in larger-scale runs.
>
>
>
> > Suitability of work: To be clear, I have no issue at all with the suitability of the work from the point of view of topic. The topic is indeed a fit for the Infrastructure track and the claimed results (10%) is also a great result for such an important/well optimized workload. My issue is with the significance/novelty of the research contributions which are mainly just the formulation of the MIP and observation regarding DP/PP misalignment. The work though has a significant amount of engineering contribution.
>
> We wish to gently point out that the MIP algorithm represents innovative system research contributions. Its motivation stems from our observation of misalignment in real-world clusters, and its effectiveness is tested on production training jobs. The 10.6% speedup of a production training job has a huge impact on cost-saving, swift productionization, and fast iteration.
>
> We appreciate your response. If you have any further concerns, please feel free to let us know. Thank you.

---

> > ### Comment · Reviewer_awuk · 2025-08-08
> > **Response**
> >
> > Thanks for the response.
> >
> > On reproducibility:
> >
> > In section 6, it is claimed that there is both a python prototype as well as a a version built on top of Kubernetes. In addition, the changes are built on top of Megatron.
> >
> > Am I correct in understanding that in order to reproduce the results all of the following are necessary?
> > 1. Megatron changes
> > 2. Either Python simulator or Kubernetes scheduling layer?
> >
> > I know that the authors have committed to open sourcing #2 which I believe to be the critical part necessary to reproduction of the results. Did I miss any other layer?
> >
> > On experiments:
> >
> > I appreciate the additional experiment on Llama 3 8B. It would be great to include such results in the main paper. However, this further illustrates my point. There seem to be a number of factors that affect performance, including scale, model type, interconnect, etc. There isn't a sufficient enough study in the main paper for the reader to understand which factors are going to matter since there are only 2 experiments run. However, I also understand the difficulty in running such large scale experiments. In any case, it is hard to understand the benefits and limitations of the model based on the experiments in the paper right now.

---

> > > ### Author Response · Authors · 2025-08-08
> > > **Response2**
> > >
> > > **Reproducibility.**
> > >
> > > > - Megatron changes
> > > > - Either Python simulator or Kubernetes scheduling layer?
> > > >
> > > > I know that the authors have committed to open sourcing #2 which I believe to be the critical part necessary to reproduction of the results. Did I miss any other layer?
> > >
> > > The reviewer is correct about the training framework and scheduling layer, and did not miss other layers. The python simulator consists of scheduling algorithms and simulation benchmarks. The scheduling algorithm is deployed as a service in Kubernetes as the scheduler.
> > >
> > >
> > > **Experiments.**
> > >
> > > > It would be great to include such results in the main paper.
> > >
> > > We will include the experiment results in the later version.
> > >
> > > > There seem to be a number of factors that affect performance, including scale, model type, interconnect, etc. There isn't a sufficient enough study in the main paper for the reader to understand which factors are going to matter since there are only 2 experiments run.
> > >
> > > We appreciate the reviewer’s insightful comments regarding these factors. We would like to clarify that these aspects have already been analyzed in Section 4 of the paper. Specifically, individual communication performance is presented in Figure 4. The effects of model scale and type are shown in Figure 5. Furthermore, the impact of network hierarchy at the spine switch level is discussed in Lines 172–177, and the influence of other network protocols is addressed in Lines 178–182.
> > >
> > > > In any case, it is hard to understand the benefits and limitations of the model based on the experiments in the paper right now.
> > >
> > > Based on the characterization of individual communication operations and end-to-end training performance in Section 4, the benefit of our algorithm is topology-aware communication alignment, enhancing the locality of GPU communication. Its effectiveness is shown in production runs in section 7.2.

---

### Official Review · Reviewer_dkH5 · 2025-07-02

**Clarity:** 3
**Significance:** 2
**Originality:** 2
**Rating:** 4
**Confidence:** 4

**Summary:**

This paper introduces a new scheduler, dubbed Arnold, for distributed training of LLMs. Arnold is responsible for determining how to assign accelerators to perform data parallelism and pipelining provided information about hardware topology and communication bandwidth. The scheduling problem is formalized as a mixed-integer program. The highlight of the empirical result shows an improvement of about 10% in throughput when training an unspecified LLM on 9600 GPUs compared to megascale, which is the current standard.

**Questions:**

- how much better is the enumeration approach in terms of throughput across various cluster configurations?

**Ethical Concerns:**

["NO or VERY MINOR ethics concerns only"]

**Final Justification:**

After reading the rebuttal and the other reviews, I have decided to keep a borderline rating but leaning towards acceptance, as there is nothing flawed in the paper as far as I can tell.
I think that if the authors include the clarifications from the reviews (e.g., why gains are significant, articulate better how Arnold adapts to various topologies), the paper might get better received.

**Limitations:**

The major limitation of this paper is the absence of connection to learning itself. This is a contribution better suited for a system conference.

**Paper Formatting Concerns:**

none that I am aware

**Quality:**

3

**Strengths And Weaknesses:**

Strengths:
+ topic is very timely
+ paper is overall clearly written

Weaknesses:
- This is a system paper, the authors made not much effort to connect this to learning. I think a system venue would be more suitable than NeurIPS.
- The empirical validation is lacking. I think throughput should have been used as a metric throughout (and not just in the last section). The time to schedule is absolutely negligible relative to the total training time.
- The reported gain when using 9600 GPUs is a bit marginal. If large runs take a month, let's say, Arnold let us save only 3 days. It is unclear to me whether this is because the scheduling is sub-optimal or mega-scale is close to optimal already.
- The paper lacks insights a bit. It would have been nice to show how different hardware topologies / cluster configurations yield / benefit from different scheduling.
- When scaling to very large models, a node may not be sufficient to host the entire model and tensor parallelism might be needed across nodes, but the paper considers only DP and PP.

---

> ### Author Rebuttal · Authors · 2025-07-30
>
> > **w1: This is a system paper, the authors made not much effort to connect this to learning. I think a system venue would be more suitable than NeurIPS.**
>
> **Scope of our paper.** The topic of "**Infrastructure** (e.g., libraries, improved implementation and scalability, distributed solutions)" is explicitly included in the call for paper statement in NeurIPS 2025. The scope of our paper falls in the **infrastructure track**, which includes improved implementation, scalability and distributed solutions. Other innovative solutions such as flash-attention, also have wide impacts on the NeurIPS community.
>
> **Connection to learning.** LLMs' learning is benefited by large-scale training, which necessitates innovative infrastructure support. In section 3 and 4, we share our insights and lessons learned from large-scale training on real-world clusters. In section 5, 6 and 7, we present a scheduling algorithm capable of handling very large-scale training jobs. We hope our work shed light on scalable training solutions and benefit the community.
>
> ---
>
> > **w2: The empirical validation is lacking. I think throughput should have been used as a metric throughout (and not just in the last section). The time to schedule is absolutely negligible relative to the total training time.**
>
> **Throughput as the metric throughout the paper.** In the simulator described in section 7.1, we use Equation (2) as the score to benchmark scheduling algorithms. This is based on the observation from Figure 4 and 5. Alternatively, we can replace the metric in the simulator by throughput via using the characterization results from section 4 (line 141-155). For example, as communication groups span over more minipods, the bandwidth decreases from the maximum bandwidth, and we use the percentage of the decrease to estimate the decrease of throughput from maximum throughput.
>
>
> setting (i) throughput (PetaFlops)
>
> | Method       | α = 0.2  | α = 0.5  | α = 0.8  |
> |--------------|--------- |--------- |--------- |
> | Best-Fit     | 1.68     | 1.63     | 1.59     |
> | Random-Fit   | 1.55     | 1.52     | 1.49     |
> | GPU-Pack     | 1.68     | 1.63     | 1.59     |
> | Topo-Aware   | 1.68     | 1.63     | 1.59     |
> | Ours         | 1.68     | 1.63     | 1.59     |
>
>
> setting (ii) throughput (PetaFlops)
>
> | Method       | α = 0.2  | α = 0.5  | α = 0.8  |
> |--------------|--------- |--------- |----------|
> | Best-Fit     | 12.57     | 12.57     | 12.57  |
> | Random-Fit   | 12.22     | 11.70     | 11.18  |
> | GPU-Pack     | 12.57     | 12.57     | 12.57  |
> | Topo-Aware   | 12.57     | 12.57     | 12.57  |
> | Ours         | 13.44     | 13.11     | 12.78  |
>
> setting (iii) throughput (PetaFlops)
>
> | Method       | α = 0.2 | α = 0.5 | α = 0.8 |
> |--------------|---------|---------|---------|
> | Best-Fit     | 46.85     | 44.85     | 42.86     |
> | Random-Fit   | 46.19     | 43.19     | 40.20     |
> | GPU-Pack     | 47.16     | 45.62     | 44.08     |
> | Topo-Aware   | 47.16     | 45.62     | 44.08     |
> | Ours         | 50.51     | 47.71     | 44.92     |
>
>
>
> **The time to schedule is not negligible.** We wish to gently point out that the time to schedule is not negligible as shown in Figure 8. The enumeration method has time complexity $O(k^m)$, i.e., for each of the $m$ nodes, we can place it in one of the $k$ minipods. This is not feasible for online scheduling. Moreover, cloud providers price Hopper GPUs at \$2.99/h. For our largest job, this is \\$28,704/hour or \\$478/min. GPUs can be idle during scheduling time, which directly leads to significant cost. Figure 8 shows that our method scales well for large job configurations and data center topology.
>
> ---
>
> > **w3: The reported gain when using 9600 GPUs is a bit marginal. If large runs take a month, let's say, Arnold let us save only 3 days. It is unclear to me whether this is because the scheduling is sub-optimal or mega-scale is close to optimal already.**
>
> **Arnold's impact on cost savings.** We wish to highlight that a 10.6% speedup has a huge impact on cost savings. If 9,600 GPUs save 3 days, this saves us \\$2,066,688, because cloud providers price Hopper GPUs at \$2.99/h. This improvement has a significant impact on cost-saving, swift productionization, and fast iteration.
>
> **MegaScale's scheduling is suboptimal.** MegaScale, which employs the bin-pack algorithm, ignores the inherent sparse communication patterns in LLM pre-training (shown in Figure 1 and 3), while Arnold's MIP algorithm specifically considers these patterns. In section 7, we demonstrate how the improved scheduling algorithm works well in both simulation and real-world environments.
>
> ---
>
> > **w4: The paper lacks insights a bit. It would have been nice to show how different hardware topologies / cluster configurations yield / benefit from different scheduling.**
>
> **Benchmarks in different setups.** We have shown benchmarks of different hardware topologies (TOR switch, spine switch), cluster configurations (small, medium, large job sizes), and different scheduling algorithms (a list of SoTA schedulers from literacture) in section 7.1 (line 290-306).
>
> **Scheduling insights.** We have shown that when scheduling small jobs in a simple topology, scheduling algorithms achieve similar scores. As the job size and scale become larger, Arnold's algorithm shows more advantageous, and it is able to scale to very large sizes, as detailed in line 321-325, section 7.1:
> >> We also observe that as $\alpha$ increases, our scheduling algorithm is closer to other baselines. This is because $\alpha$ controls the affinity of the DP group, and if $\alpha$ is $1$, the objective function reduces to a bin-packing formulation and therefore has no difference from other bin-packing algorithms. In practice, we would not set $\alpha$ greater than $0.5$ as observed from our characterization results. As a result, our algorithm usually scores higher than other baselines.
>
>
> ---
>
> > **w5: When scaling to very large models, a node may not be sufficient to host the entire model and tensor parallelism might be needed across nodes, but the paper considers only DP and PP.**
>
> **Clarification.** Our model is not hosted by one node. As discussed in section 7.2 (line 354-358), our training systems have trained LLMs with more than 400 billion parameters, and it is sharded by TP and PP across multiple nodes.
>
> **Why only PP and DP are considered.** TP must stay within high-bandwidth low-latency network domain (e.g. intra-node NVLink), as Megatron-LM [1] has in-depth experiments to show that. PP can scale out to inter-node connections. Therefore, to scale even larger than our models, more PP stages can be added to accommodate the memory pressure.
>
> [1] Deepak Narayanan, Mohammad Shoeybi, Jared Casper, Patrick LeGresley, Mostofa Patwary, Vijay Korthikanti, Dmitri Vainbrand, Prethvi Kashinkunti, Julie Bernauer, Bryan Catanzaro, Amar Phanishayee, and Matei Zaharia. Efficient large-scale language model training on gpu clusters using megatron-lm. In SC21: International Conference for High Performance Computing, Networking, Storage and Analysis, pages 1–14, 2021.
>
> ---
>
> > **q1: how much better is the enumeration approach in terms of throughput across various cluster configurations?**
>
> **Compare with enumeration approach on setting (i).** We add additional experiments on setting (i) (line 302) comparing the throughput of the enumeration approach with our scheduling algorithm:
>
> setting (i) throughput (PetaFlops)
>
> | Method       | α = 0.2  | α = 0.5  | α = 0.8  |
> |--------------|--------- |--------- |--------- |
> | Best-Fit     | 1.68     | 1.63     | 1.59     |
> | Random-Fit   | 1.55     | 1.52     | 1.49     |
> | GPU-Pack     | 1.68     | 1.63     | 1.59     |
> | Topo-Aware   | 1.68     | 1.63     | 1.59     |
> | Ours         | 1.68     | 1.63     | 1.59     |
> | Enumeration  | 1.68     | 1.63     | 1.59     |
>
> As shown in the table, our scheduling algorithm can match the performance of the enumeration method with much shorter scheduling time.
>
> **Compare on other settings.** For setting (ii) and (iii), the algorithm has complexities of $O(5^{96})$ and $O(11^{368})$. They are computationally intractable.

---

> > ### Comment · Reviewer_dkH5 · 2025-08-02
> > **thank you**
> >
> > I thank the authors for their response.
> > I found it only partially satisfactory. I appreciate better how the 10% gain is important, but I still have concerns regarding the general lack of insights. For instance:
> > - could you visualize what the scheduling does in the two different topologies considered or could you add a third one where it does something quite different than megascale?
> > - or what insights do we gain by seeing how Arnold better optimizes larger scales?
> > - could you pick a less trivial example where to report the oracle (as in the example provided all methods perform equally well to it)?
> >
> > Despite everything, I reckon the engineering achievement and the significance for our field. As there is nothing flawed, I've raised my ratings to borderline accept.

---

> > > ### Author Response · Authors · 2025-08-04
> > >
> > > We thank the reviewer for the acknowledgement.
> > >
> > > ---
> > >
> > > **q1. could you visualize what the scheduling does in the two different topologies considered or could you add a third one where it does something quite different than megascale?**
> > >
> > > **Visualization.** We have an internal tool to visualize the communication matrix as formulated in Equation (1) and Figure 12, where nodes are painted with distinct colors for each minipods. Although we are not able to show the exact scheduling due to the scale and confidentiality, we observe a simple bin-packing strategy spreads nodes easily across minipods. For example:
> > >
> > > | Comm. group | PP0   | PP1   | PP2   |
> > > |-------------|-------|-------|-------|
> > > | DP0         | ◯ (rank0)    | ◼ (rank1)    | ▲ (rank2)    |
> > > | DP1         | ◼ (rank3)    | ◯ (rank4)    | ◆ (rank5)    |
> > >
> > > Arnold's scheduling aligns the node better with the topology, e.g.,
> > >
> > > | Comm. group | PP0   | PP1   | PP2   |
> > > |-------------|-------|-------|-------|
> > > | DP0         | ◼ (rank0)    |   ◯ (rank1)  | ▲ (rank2)    |
> > > | DP1         | ◼ (rank3)   | ◯ (rank4)    | ▲ (rank5)     |
> > >
> > > Essentially, at scheduling time, the schedulers would have to be aware of the communication matrix to make a good scheduling decision.
> > >
> > > **q2. or what insights do we gain by seeing how Arnold better optimizes larger scales?**
> > >
> > > Our insight is shown by Figure 10 and 15, where they show the interplay between communication and computation kernels, and how scheduling can make a difference.
> > >
> > > Computation is much faster than communication for GPUs (TeraFlops vs GB/s). Thus, computation kernels are blocked by the dependent communication kernels. At larger scale, the training is bottlenecked by communication, and the optimized topology benefits GPU utilizes resources for communication, which is more pronounced when both computation and communication sizes increase.
> > >
> > >
> > > **q3. could you pick a less trivial example where to report the oracle (as in the example provided all methods perform equally well to it)?**
> > >
> > > As explained, the $O(k^m)$ complexity makes the enumeration method unfeasible for jobs and configuration larger than the simple setting. We manage to construct an artificial simple topology to show the difference between the enumeration approach and our MIP. This artificial topology is unlikely in practice, because we force to spread the available node (the total scheduling space) across several minipods.
> > >
> > > | Minipods       | Number of Nodes  |
> > > |--------------|--------- |
> > > | 1     | 1     |
> > > | 2  | 1     |
> > > | 3     | 1     |
> > > | 4  | 2     |
> > > | 5     | 2     |
> > > | 6  | 2     |
> > > | 7  | 3     |
> > >
> > > Estimated throughput (PetaFlops):
> > >
> > > | Method       | α = 0.2  | α = 0.5  | α = 0.8  |
> > > |--------------|--------- |--------- |--------- |
> > > | Ours         | 1.52     |  1.44    |   1.37   |
> > > | Enumeration  | 1.53     |   1.48   |   1.44   |

---

### Note · Authors · 2025-08-12

**Overall merits**: we thank all reviewers for their valuable comments. The reviews acknowledge our paper as solid ML system work, with noted strengths including the problem being well-motivated, the novel scheduling algorithm being significant for LLM training, and the system being validated in production environment. We believe the lessons learned from large-scale training and our proposed solution validated by production runs will be impactful for the community.


**Main concerns in discussion:**

- Alternative metrics in simulation. We have performed simulation experiments using the percentage of the decrease to estimate the decrease of throughput from maximum throughput. The results show our algorithm also works well using the alternative metric.

- The MIP formulation does not provide sufficient detail in its explanation. In the rebuttal, we have provided a more thorough explanation along with a simple example to clarify the formulation. These details will be incorporated into the final version.

- The production training is based on our proprietary LLMs. To further validate our approach, we have added new results using LLaMA-3 8B, demonstrating that our scheduling algorithm also performs well for open-source models and commodity networks (RoCE). Moreover, the discussions in Sections 3 and 4 are rooted in the concept of data locality, a fundamental principle in computer science. We have conducted comprehensive experiments on both individual communication operations and end-to-end training performance, covering diverse model configurations and network topologies. We believe the resulting insights are broadly applicable.

We hope our rebuttal has addressed the concerns raised. We would greatly appreciate it if the AC and reviewers could consider the overall merits of our paper together with the clarifications provided in our rebuttal.

---

### Decision · Program_Chairs · 2025-09-17

**Decision:**

Accept (poster)

**Comment:**

This paper presents a new scheduling system for LLM pre-training. It identifies the mismatch between physical network topology and LLM pre-training jobs and effectively align the communication patterns with the network topology to optimize communication efficiency. Large-scale simulations and end-to-end physical experiments demonstrate the effectiveness of the proposed system.

The reviewers agree that this MLSys work is well written, addressing a timely topic, and the evaluation validates the superiority in terms of training efficiency. The majority of reviewer’ comments have been addressed in the authors’ responses. Based on these, this paper is recommended for acceptance. The authors are suggested to include the important experiment results and discussion in the responses to the paper. Besides, the authors should add more clarifications about the reviewers’ remaining points, such as highlighting the technical novelty and insights, and adding more clarifications about the system design and empirical results.